# The Renin–Angiotensin System and Cardiovascular–Kidney–Metabolic Syndrome: Focus on Early-Life Programming

**DOI:** 10.3390/ijms25063298

**Published:** 2024-03-14

**Authors:** You-Lin Tain, Chien-Ning Hsu

**Affiliations:** 1Division of Pediatric Nephrology, Kaohsiung Chang Gung Memorial Hospital, Kaohsiung 833, Taiwan; tainyl@cgmh.org.tw; 2College of Medicine, Chang Gung University, Taoyuan 333, Taiwan; 3Institute for Translational Research in Biomedicine, Kaohsiung Chang Gung Memorial Hospital, Kaohsiung 833, Taiwan; 4Department of Pharmacy, Kaohsiung Chang Gung Memorial Hospital, Kaohsiung 833, Taiwan; 5School of Pharmacy, Kaohsiung Medical University, Kaohsiung 807, Taiwan

**Keywords:** cardiovascular disease, chronic kidney disease, metabolic syndrome, renin–angiotensin system, obesity, hypertension, developmental origins of health and disease (DOHaD), angiotensin-converting enzyme

## Abstract

The identification of pathological links among metabolic disorders, kidney ailments, and cardiovascular conditions has given rise to the concept of cardiovascular–kidney–metabolic (CKM) syndrome. Emerging prenatal risk factors seem to increase the likelihood of CKM syndrome across an individual’s lifespan. The renin–angiotensin system (RAS) plays a crucial role in maternal–fetal health and maintaining homeostasis in cardiovascular, metabolic, and kidney functions. This review consolidates current preclinical evidence detailing how dysregulation of the RAS during pregnancy and lactation leads to CKM characteristics in offspring, elucidating the underlying mechanisms. The multi-organ effects of RAS, influencing fetal programming and triggering CKM traits in offspring, suggest it as a promising reprogramming strategy. Additionally, we present an overview of interventions targeting the RAS to prevent CKM traits. This comprehensive review of the potential role of the RAS in the early-life programming of CKM syndrome aims to expedite the clinical translation process, ultimately enhancing outcomes in cardiovascular–kidney–metabolic health.

## 1. Introduction

The growing recognition of pathological links among metabolic risk factors such as obesity and diabetes, cardiovascular disease (CVD), and chronic kidney disease (CKD) has given rise to the conceptualization of cardiovascular–kidney–metabolic (CKM) syndrome [1]. In its 2023 Scientific Statement, the American Heart Association, for the first time, defined CKM syndrome as a systemic disorder characterized by intricate pathophysiological interactions among metabolic risk factors, CKD, and the cardiovascular system. These interactions give rise to multiorgan dysfunction and elevate the risk of adverse cardiovascular and renal outcomes [1]. CKM syndrome has been categorized into four separate stages, ranging from stage 0 to stage 4. These stages are believed to encompass varying degrees of advancement and intensity within the complex spectrum of this disorder. Different critical elements emerge at various stages, playing a role in the nuanced development and severity observed across the intricate spectrum of CKM syndrome.

An estimated 40% of adults in the United States are thought to be impacted by CKM syndrome [2]. Given it results in multi-organ dysfunction, there is a significant global burden on compromised cardiovascular–kidney–metabolic health. While managing this syndrome is recommended through a holistic approach that addresses the entire syndrome rather than individual diseases [2], there is still a lack of therapeutic guidelines. It is important to highlight that prioritizing early prevention has the potential to mitigate the burdens linked with CKM syndrome. Acknowledging the interconnections among CKM diseases is vital for embracing a more comprehensive approach to CKM care, surpassing the isolated treatment of individual conditions. This broader perspective shows promise in improving global health outcomes in the future.

It is now widely accepted that the risks of many chronic diseases in adulthood may have their origins in early life [3,4]. Experiencing a suboptimal intrauterine environment during development leads to enduring adverse effects on both structure and function, as well as on compensatory mechanisms, a phenomenon referred to as developmental programming or the “developmental origins of health and disease” (DOHaD) [5]. The DOHaD theory proposes a link between early life programming and several recognized components of CKM syndrome, covering metabolic disease [6], chronic kidney disease (CKD) [7], CVD [8], hypertension [9], and obesity [10]. Conversely, this theory prompts a theoretical transition in therapeutic strategies, shifting the focus from addressing diseases in adulthood to intervening at an earlier stage—specifically, engaging in reprogramming efforts with the aim of potentially reversing disease processes before they manifest clinically [11,12].

Various molecular mechanisms associated with the developmental programming of CKM syndrome have been explored. These encompass renin–angiotensin system (RAS) dysgenesis, nitric oxide (NO) deficiency, epigenetic regulation, oxidative stress, disruptions in nutrient-sensing signals, a low nephron number, and gut microbiota dysbiosis [11,12,13,14,15,16,17]. Among these suggested mechanisms, the RAS serves as a central hub intricately linked with other factors in influencing the adverse programming processes.

The RAS operates as a hormonal cascade, commencing with the expression of angiotensinogen (AGT), which is converted into angiotensin (Ang) I by the renin enzyme. Subsequently, Ang I is cleaved into Ang II by angiotensin-converting enzyme (ACE) [18]. The RAS plays a pivotal role in orchestrating various physiological functions within the cardiovascular system, kidneys, and metabolic homeostasis [19,20]. Conversely, several pathological effects, such as vasoconstriction and cell proliferation, are frequently induced by Ang II through the activation of the classical RAS pathway, comprising ACE, Ang II, and the Ang II type 1 receptor (AT1R) in CKM syndrome. These effects contribute to conditions such as hypertension, CKD, obesity, liver steatosis, and diabetes [20,21,22,23,24]. On the flip side, the non-classical RAS pathway, involving the ACE2-ANG-(1-7)-MAS receptor axis, serves to counterbalance the detrimental effects of Ang II signaling [25].

Within this framework, the RAS has surfaced as a pivotal focal point for comprehending and averting CKM syndrome with developmental origins. Inhibition of the classical RAS or stimulation of the non-classical RAS serves as the rationale for existing cardioprotective, antihypertensive, renoprotective, and anti-obesity therapies [18,25,26,27,28]. While limited data are available on whether early targeting of the RAS can prevent offspring’s CKM syndrome, the objective of this review is to investigate the mechanistic link between the RAS and the developmental programming of CKM syndrome. Utilizing scientific databases such as SCOPUS, Embase, MEDLINE, and the Cochrane Library, we sought to summarize the relationship among the RAS, developmental programming, and CKM syndrome. This involved addressing the molecular mechanisms and identifying potential RAS-targeted reprogramming interventions for the prevention of CKM syndrome. The search encompassed keywords and their combinations such as “hypertension”, “chronic kidney disease”, “obesity”, “metabolic syndrome”, “diabetes”, “hyperlipidemia”, “cardiovascular disease”, “developmental programming”, “DOHaD”, “offspring”, “mother”, “nephron”, “pregnancy”, “gestation”, “lactation”, “progeny”, “reprogramming”, “prorenin receptor”, “aldosterone”, “mineralocorticoid receptor”, “angiotensinogen”, “angiotensin-converting enzyme”, “renin”, and “angiotensin”. Supplementary investigations were chosen and assessed utilizing pertinent references found in eligible papers, with the final search conducted on 30 January 2024.

## 2. Systemic and Local RAS

Initiating the RAS cascade is renin, with its precursor, prorenin (406 amino acids) [29], undergoing proteolytic cleavage exclusively in the kidney, where both renin and prorenin are secreted into the circulation [30]. Renin (340 amino acids), acting as a hormone [31], binds to (pro)renin receptor (PRR), encoded by Atp6ap2 in three forms [32]. The interaction of circulating renin and prorenin with PRR triggers Ang II-independent signaling cascades, initiating local Ang II generation.

The RAS substrate AGT, released from the liver, is cleaved by renin to produce Ang I. ACE further cleaves Ang I, leading to Ang II formation in various tissues [33]. While AT1R stimulation by Ang II increases sodium reabsorption and raises BP, AT2R mediates vasodilation and lowers BP [34]. Ang II also promotes lipogenesis [35], increases adipose tissue mass, and stimulates the adrenal gland cortex to secrete aldosterone, maintaining sodium–potassium homeostasis. The renal RAS, with the highest tissue concentrations of ANG II, involves the metabolism of Ang II to Ang III and Ang IV [36].

ACE2 converts Ang II to Ang-(1-7) or Ang I to Ang-(1-9). Ang-(1-7), mediated by the MAS receptor, induces natriuretic and diuretic effects, promoting vasodilation [37]. Neprilysin (NEP) facilitates the conversion of Ang I to Ang-(1-7), with subsequent metabolic processing generating Ang-(2-7) and Ang-(3-7).

Distinguishing between the local and systemic RAS poses challenges due to extensive overlap [38]. The local adipose RAS, expressed in adipose tissues, modulates processes such as adipogenesis, lipogenesis, lipolysis, and inflammation [39]. The kidney houses a potent local vascular RAS for independent renal vascularization. A distinct urinary RAS in the kidney coordinates sodium reabsorption [40].

A comprehensive understanding of the RAS peptide network’s influence on fetal programming requires recognizing the collaborative or opposing nature of different peptides. Pharmacological modifications induce compensatory adjustments in RAS enzymes, necessitating further research to unravel the complexities of this network and its impact on fetal programming, as illustrated in Figure 1.

## 3. CKM Syndrome Is Causal of RAS Perturbation

### 3.1. Cardiovascular Disease and Hypertension

For a long time, the endothelium was considered involved in the regulation of vascular homeostasis [41]. Vascular endothelial function is primarily maintained by the balanced production of endothelial relaxing factors, namely, nitric oxide (NO), as well as endothelial contractile factors such as Ang II or superoxide anion [42]. Endothelial dysfunction is characterized by a vasoconstrictive, proadherent, prothrombotic, proliferative, and proinflammatory environment that leads to atherosclerosis, which is the initial event in the development of CVD [42]. Specifically, in hypertension, endothelial dysfunction leading to decreased NO availability impairs endothelium-dependent vasodilation [43].

Endothelial dysfunction can occur by the activation of PRR and resultant high Ang II activity [44]. Renin may interact with PRR to be of relevance in CVD in many ways [45], covering the enhancement of the RAS by catalyzing Ang I production [46], activation of mitogen-activated protein kinase (MAPK) signaling pathways [47], association with V-ATPase implicating a non-RAS-related function [48], and regulation of the Wnt/β-catenin pathway [49]. As reviewed elsewhere, activation of PRR in cardiomyocytes may contribute to myocardial ischemia/reperfusion injury, cardiac hypertrophy, diabetic cardiomyopathy, salt-induced cardiac damage, and heart failure [49].

Recognized as a significant signaling constituent of the classical effects of Ang II is Ang II-derived superoxide [50]. The major source of superoxide that impacts the cardiovascular system is reduced nicotinamide-adenine dinucleotide phosphate (NADPH) oxidase. The resulting NADPH oxidase-derived superoxide mediates many of the actions of Ang II, including constriction of vascular smooth muscles, endothelial dysfunction, increased BP, vascular remodeling, and sodium retention [50]. In addition, the activation of AT1R via Ang II induces vasoconstriction and increases the activity of the sympathetic nervous system. These increase the BP and aldosterone secretion, and generate cardiac hypertrophy and fibrosis [51].

### 3.2. Kidney Disease

During nephrogenesis, components of the RAS exhibit high expression and have crucial roles in orchestrating proper renal structure and physiological function [52]. In the case of rats, all RAS components are detectable in embryonic kidneys from gestational days 12 to 17, with higher levels observed in fetuses and newborn rats compared to adults [53]. In human studies, drugs that interfere with the RAS, such as ACE inhibitors (ACEIs) or angiotensin receptor blockers (ARBs), have been intentionally steered clear of in pregnant women. This cautious approach stems from the perceived risk of renal malformations and ACEI/ARB fetopathy [54]. Animals lacking RAS genes exhibit significant renal maldevelopment [55,56]. Blockade of the RAS during the nephrogenesis stage leads to a reduced number of nephrons and hypertension in adulthood [57].

Correlating with the presence and severity of the underlying kidney disease is the expression of RAS components in human kidney biopsies [58,59]. Likewise, a rise in classical RAS components within the renal system has been noted in several animal models of CKD, including streptozotocin (STZ)-induced diabetic nephropathy [60], five/six ablation/infarction [61], and adenine-induced CKD [62]. In the kidneys, Ang II is generated in notably high concentrations within the interstitial space. Local production of Ang II can profoundly impact renal function by modifying glomerular hemodynamics, reducing sodium excretion, and constricting small arterioles [63]. Furthermore, excessive activity of the RAS directs proinflammatory and profibrotic factors to harm the kidneys [64], while the inhibition of the RAS has demonstrated efficacy in ameliorating renal fibrosis [65].

### 3.3. Obesity

Most constituents of the RAS have been observed to be expressed in adipose tissue [66]. This localized adipose RAS plays crucial autocrine/paracrine roles in regulating processes such as lipogenesis, adipogenesis, lipolysis, and inflammation in both systemic and adipose tissue contexts [66].

In cases of obesity, the classical RAS is activated, leading to increased lipogenesis, decreased lipolysis, and the promotion of adipocyte growth and differentiation. These processes are closely linked to obesity, insulin resistance, and inflammation. The elevated adipose mass subsequently contributes to further disruptions in BP, glucose, and lipid levels. Consequently, obesity becomes a risk factor for the development of type 2 diabetes mellitus, CVD, and kidney disease, creating a cycle of pathological interconnections in CKM syndrome [67]. Conversely, heightened activation of the non-classical RAS axis has the potential to improve lipid profiles and insulin resistance, mitigate inflammation, and reduce obesity [68].

### 3.4. Diabetes

Ang II-induced increments in oxidative stress, inflammation, and free fatty acid levels contribute to beta-cell dysfunction in diabetes [69]. Various organs play a role in the regulation of glucose homeostasis, including the pancreas, adipose tissue, skeletal muscle, and liver. Significantly, a local RAS has been identified in these organs, and its activation has been implicated in the pathology of diabetes [70].

Moreover, RAS activation appears to enhance the effects of other pathogenic pathways, including glucotoxicity, lipotoxicity, and advanced glycation, leading to hyperglycemia and insulin resistance [70]. In experimental models of type 2 diabetes, the inhibition of the classical RAS or the activation of the non-classical RAS demonstrates improvements in islet structure and function [71,72,73].

### 3.5. Dyslipidemia and Fatty Liver

Non-alcoholic fatty liver disease (NAFLD) is a consequence of metabolic disorders, including obesity, insulin resistance, and metabolic syndrome. Dyslipidemia plays a crucial role in the development of NAFLD. The presence of free fatty acids and lipid metabolites within hepatocytes disrupts insulin-triggered cell signaling, leading to the onset of NAFLD [74].

Hyperglycemia, hypercholesterolemia, and insulin resistance can upregulate components of the RAS [75,76]. RAS activation and the expression of its elements in liver tissues are drivers of hepatic fatty acid metabolism, inflammation, and fibrosis [77]. Conversely, several studies indicate that ARBs exert beneficial effects on dyslipidemia [74] as well as NAFLD [78].

As outlined above in this review, intricate associations between the RAS and CVD, kidney disease, and metabolic disorders are evident. Early-life exposure to unfavorable environmental factors may trigger abnormal RAS activation, culminating in the onset of CKM syndrome in later stages of life (Figure 2).

## 4. The RAS in Pregnancy

Significantly influencing cardiovascular and kidney development in pregnant women and the fetus is the RAS. Throughout a healthy pregnancy, blood pressure tends to stay lower, while plasma renin activity and aldosterone levels remain elevated until late in pregnancy, at which point BP increases [79]. Elevated aldosterone concentrations, induced by Ang II, directly stimulate renal sodium and fluid retention, thereby enhancing the blood volume. Predominantly present in the fetal circulation during pregnancy is ACE, originating from endothelial cells. Its primary functions in this context include supporting angiogenesis and ensuring the maintenance of fetal perfusion [80]. Pregnancy also triggers the activation of the non-classical RAS pathway to counterbalance the heightened Ang II signaling pathway. This adaptation contributes to maternal hemodynamic adjustments, placental functions, and vascular remodeling [81]. In the fetal kidney, the RAS plays a crucial role in ensuring proper kidney structure formation and physiological function [52,53].

In pregnancies facing challenges, the RAS has the potential to negatively impact the cardiovascular and kidney health of both the fetus and the mother. Elevated plasma levels of PRR during pregnancy complications such as intrauterine growth restriction (IUGR), preeclampsia, and gestational diabetes mellitus have been observed [82]. In a rat model of placental insufficiency leading to IUGR, there is a correlation with reduced intrarenal RAS activity in neonatal rats [83].

Preeclampsia in women is associated with increased circulating levels of autoantibodies targeting AT1R, contributing to vasoconstriction, hypertension, and heightened coagulation [84]. RAS activation is also linked to adverse outcomes such as preterm birth [85], gestational diabetes [86], and pregnancy-induced hypertension [87]. Conversely, reduced levels of angiotensin-(1-7) in pregnant women have been observed in conditions like preeclampsia [88], preterm birth [89], and gestational diabetes [90]. Consequently, these RAS components may instigate secondary alterations in the neurohormonal regulation of cardiovascular and kidney function, potentially programming hypertension, kidney disease, and cardiovascular disease (CVD) [91]. Nevertheless, the specific timing of these RAS changes and their significance in the later development of CKM syndrome remain largely unclear.

## 5. RAS-Related Programming in Animal Models

Table 1 provides an overview of animal models that manifest at least two components of CKM syndrome in their offspring, particularly those linked to aberrant RAS alterations [92,93,94,95,96,97,98,99,100,101,102,103,104,105,106,107,108,109,110,111,112,113,114,115,116,117,118,119,120,121,122,123,124,125,126,127,128,129,130,131,132,133]. Various animal models employing diverse environmental stressors have been developed to investigate specific aspects of CKM syndrome, including hypertension [12,134], metabolic syndrome [135], kidney disease [13], and CVD [15], as discussed in previous reviews. Despite the focus on inducing distinct components of CKM syndrome in these models, none of them successfully replicate the complete set of features associated with CKM syndrome.

### 5.1. Maternal Nutritional Imbalance

Highlighted among the most commonly established are models of maternal nutritional imbalance (Table 1). These models involve specific nutritional manipulations during pregnancy and/or lactation, including caloric restriction, protein restriction, high-fructose consumption, and high-fat intake. Because human nephrogenesis is complete at term birth, most preclinical models target equivalent windows in animals in which kidney development continues after birth. For example, in rodents, kidney development persists for 1–2 weeks after birth. This approach allows researchers to explore the impact of exposures during organogenesis on the long-term health of the kidneys and cardiovascular system.

Protein restriction during pregnancy leads to hypertension, insulin resistance, and kidney disease in adult offspring, which are related to RAS programming effects. Increased renal AT1R expression and decreased AT2R expression were found in 4-week-old progeny born to dams that received a low-protein diet [96].

Maternal high-fructose diet programs increased BP and increased renal renin and brain AT1R expression in male rat offspring [101,102,103]. In a particular investigation, it was observed that a maternal high-fructose diet could lead to the multigenerational activation of the RAS [102]. The study revealed a significant elevation in BP among first- and second-generation offspring compared to the control group, although this effect was not observed in the third and fourth generations. The third-generation offspring exhibited the highest increases in serum renin, Ang II, and aldosterone levels. Additionally, this dietary pattern resulted in heightened mRNA expression of RAS-related genes in the kidneys from the first to third generations of rat offspring [102].

High-fat diets have consistently been shown in animal models to be linked to the emergence of obesity and related diseases [136,137]. From animal models, current evidence has emerged indicating that progeny exposed to a maternal high-fat diet manifest various characteristics of CKM syndrome [104,105,106,107], including obesity, hypertension, insulin resistance, dyslipidemia, and kidney disease. Offspring hypertension, primed by a maternal high-fat diet, is associated with the aberrant activation of the classical RAS. This is manifested by elevated renal mRNA expression of AGT and ACE, along with an increased protein level of AT1R [104]. Another study demonstrated that in 16-week-old male offspring born to dams exposed to a high-fat diet a notable decrease in the renal level of Ang-(1-7) was observed [105].

### 5.2. Maternal Illnesses and Conditions

During gestation, maternal illnesses and conditions can have significant implications for fetal programming, elevating the risk of offspring developing CKM syndrome. Consequently, animal models replicating maternal illnesses and conditions have been established to investigate different facets of CKM syndrome, including hypertension, obesity, insulin resistance, dyslipidemia, and kidney disease (refer to Table 1). The spectrum of maternal illnesses and conditions encompass maternal diabetes [108,109,110], CKD [111,112], uteroplacental insufficiency [113,114,115], and maternal hypoxia [116,117].

Offspring born to streptozotocin (STZ)-treated diabetic mother rats displayed hypertension, obesity, insulin resistance, dyslipidemia, and kidney disease [108,109,110]. Maternal diabetes led to an upregulation of ACE and AT1R, coupled with a downregulation of ACE2 expression in the kidneys of the offspring [108]. Furthermore, maternal diabetes resulted in hypertension in the offspring, accompanied by an elevation in ACE activity [109].

The adult offspring of mothers with adenine-induced CKD exhibited hypertension and renal hypertrophy. These effects were correlated with an upregulation of the renal gene expression of AGT, renin, PRR, ACE, and AT1R, along with the downregulation of AT2R and MAS [111,112]. Uteroplacental insufficiency in rats serves as a model of IUGR and subsequent developmental programming of hypertension, dyslipidemia, insulin resistance, and kidney disease in the offspring [83,113,114,115]. Offspring hypertension in this model is linked to Ang II-dependent hypertension, with augmented renal ACE activity and AGT and ACE mRNA expression in adult progeny [83]. Maternal hypoxia is another model causing offspring CKM, which is linked to the programming of BP responses to Ang II [116,117].

### 5.3. Drug and Chemical Exposures

Various drug and chemical exposures can induce offspring CMK phenotypes mediated by the RAS. Prenatal dexamethasone exposure upregulates RAS components and results in obesity, hypertension, insulin resistance, and kidney disease in adult rat progeny [92,118,119,120,121]. Antenatal glucocorticoid exposure causes offspring hypertension, coinciding with the upregulation of renin, PRR, ACE, and AT1R expression [92,118]. Another investigation revealed that prenatal exposure to dexamethasone leads to dysfunction of β-cells and glucose intolerance attributed to the suppression of ACE2 expression [119].

In addition, prenatal exposure to nicotine results in hypertension, hyperlipidemia, steatosis, and kidney disease, all of which are traits associated with CKM in adult offspring [122,123,124,125]. The sensitization of male rat offspring to the hypertensive effects of Ang II due to antenatal nicotine exposure is reported. Another example is ethanol exposure. Prenatal ethanol exposure can induce kidney disease in adult rat offspring, coinciding with the aberrant RAS [126]. Increased gene expression of ACE and AT1R was noted with prenatal ethanol exposure, whereas it led to a reduction in the expression of AT2R, ACE2, and MAS [126].

Moreover, Table 1 illustrates that prenatal exposure to 2,3,7,8-tetrachlorodibenzo-p-dioxin (TCDD) or di(2-ethylhexyl) phthalate (DEHP) induces CKM phenotypes in adult rat offspring [128,129,130,131,132,133]. In the maternal TCDD exposure model, offspring hypertension is associated with increased renal AT1R expression [128]. In the maternal DEHP exposure model, impaired kidney development and adult kidney disease have been attributed to the inhibition of the RAS [130].

In summary, a diverse range of maternal insults is employed in animal models to investigate the programming of the RAS and its subsequent impact on the cardiovascular, kidney, and metabolic health of offspring. Collectively, these studies underscore the various mechanisms that can modify the RAS. Importantly, they emphasize the need to target the RAS for reprogramming interventions, a crucial step in the early prevention of CKM syndrome.

## 6. Targeting the RAS as a Reprogramming Strategy

To date, strategies for early-life interventions aimed at mitigating the mechanisms associated with DOHaD range from avoiding risk factors, implementing nutritional interventions, and employing pharmacological therapies to making lifestyle modifications [138,139,140]. Given the substantial progress in our comprehension of the mechanisms governing RAS programming in offspring in recent years, there is an imperative to devise innovative reprogramming strategies targeting the RAS for the prevention of CKM syndrome. Indicated currently for the treatment of hypertension, CVD, and CKD are ACEIs and ARBs. Their use has been associated with improved survival and significant cardiovascular and kidney benefits in high-risk patients [18]. Nevertheless, limited information is available regarding their reprogramming effects on CKM syndrome. Table 2 compiles the literature detailing the utilization of RAS-targeted interventions for CKM phenotypes, specifically focusing on interventions initiated before the clinical phenotype manifests.

Presently, several RAS-targeted interventions have been examined in animal models of CKM syndrome, including renin inhibitors [101,141,142,143], ACEIs [143,144,145,146,147,148], ARBs [141,149,150,151], AT1R antisense [152], and ACE2 activators [153]. The predominant protective effects of a variety of RAS-targeted interventions against CKM traits primarily encompass hypertension, succeeded by concerns such as kidney disease [148,151,153] and CVD [143,151]. While the inhibition of the RAS has demonstrated advantages in addressing other aspects of CKM syndrome, including obesity, liver steatosis, and diabetes [21,22,23,24], its impact on the reprogramming of these phenotypes remains uncertain.

Investigations into the reprogramming effects of RAS-based treatments have been conducted in rats aged between 10 and 30 weeks, approximately aligning with human ages from childhood to young adulthood [154]. However, the majority of these studies have predominantly concentrated on male subjects and have not delved into the exploration of different dosage levels. Further research is essential to clarify whether these observed effects manifest in a dose- or sex-dependent manner.

The proposition of early inhibition of the classical RAS axis aims to reprogram the aberrant activated RAS, thereby preventing CKM syndrome. In rodents, kidney development is entirely completed by postnatal weeks 1–2, and cardiomyocytes seldom reenter or advance through the cell cycle after postnatal day 9. Consequently, suitable therapeutic windows entail the initiation of treatments in juvenile offspring, commencing as early as postnatal 2 weeks in most rodent models. As listed in Table 2, typical therapeutic periods involve treating juvenile offspring with aliskiren [101,142], captopril [144], or losartan [149,150] between the ages of 2 and 4 weeks. This aims to mitigate adverse programming processes without compromising kidney development.

Currently, aliskiren holds the distinction of being the inaugural renin inhibitor sanctioned for the treatment of hypertension. Two studies indicate that when administered to offspring aged 2–4 weeks, aliskiren can prevent hypertension in adults whose mothers were fed a high-fructose diet [101] or subjected to caloric restriction [142]. Another study investigated the potential of aliskiren and lisinopril, administered between postnatal days 12 and 18, to thwart hypertension and diabetic retinopathy in a diabetic (mRen-2)27 rat model [143]. Aliskiren demonstrated superior retinal protection compared to lisinopril, although lisinopril exhibited better normalization of BP than aliskiren [143]. However, aliskiren falls short in impeding the interaction between the PRR and its ligand. Despite the reported positive effects of PRR inhibitory peptides such as the handle region peptide and PRO20 [155,156] in animal models, questions persist regarding their specificity and efficacy [157]. There is optimism that the development of a specific non-peptide inhibitor for PRR could yield favorable outcomes in (pro)renin–PRR inhibition in the imminent future.

As the predominant animal model for essential hypertension and its associated metabolic disturbances, the spontaneously hypertensive rat (SHR) is utilized [158]. The early post-weaning administration of ACE inhibitors, such as captopril [145] or perindopril [148], for a duration of 3 weeks has demonstrated efficacy in preventing the onset of hypertension in adult SHRs. Similarly, the early use of captopril [144] or enalapril [146] has exhibited beneficial effects on countering hypertension in offspring programmed by maternal protein restriction.

Losartan stands out as the sole ARB investigated in programmed CKM syndrome (Table 2). In a rat model of uteroplacental insufficiency, administering losartan between 5 and 8 weeks of age has been found to protect adult offspring from hypertension, vascular dysfunction, and kidney disease [151]. Another study highlighted the preventive potential of early treatment with AT1R antisense against hypertension in SHRs [152]. It is noteworthy that the initiation of AT1R antisense delivery occurred at postnatal day 5 [152], and its impact on the nephron number remains unexplored.

Pharmacological interventions have traditionally focused on inhibiting the classical RAS. However, with the identification of the alternative RAS, researchers have explored alternative strategies to activate this non-classical RAS with limited success until recently [159]. Surprisingly, little attention has been directed toward applying this approach to programmed CKM syndrome. Highlighted in Table 2 is the finding that only two studies have documented the administration of diminazene aceturate (DIZE), a potential ACE2 activator, or ANG-(1-7) during pregnancy. This administration was shown to alleviate hypertension and renal fibrosis in adult SHR offspring [153].

Despite the therapeutic potential of activating the non-classical RAS axis in various diseases, further investigations are warranted to delineate its reprogramming effects on CKM programming. A significant gap in the literature lies in gaining a deeper understanding of the pivotal components of the RAS for a targeted approach and determining the optimal therapeutic window to prevent CKM syndrome with developmental origins.

## 7. Conclusions and Future Directions

While the dysregulation of the RAS is recognized as one of the factors contributing to the programming of components within CKM syndrome, significant gaps persist in the field, primarily due to methodological constraints and a lack of consensus that has impeded translation into clinical practice.

A major unresolved issue is the scarcity of studies undertaking a comprehensive analysis simultaneously quantifying the expression and activity of the entire spectrum of RAS components in experimental settings. Given the intricate nature of RAS signaling, relying on the analysis of isolated components may lead to a misinterpretation of the system’s functional status.

The utilization of drugs to modulate the RAS is well-established in clinical practice, although it is still emerging in the field of fetal programming. This review presents data from animal models showcasing various RAS-based therapies that demonstrate positive effects on CKM programming, including renin inhibitors, ACEIs, ARBs, AT1R antisense, and ACE2 activators. However, the reprogramming effects in response to early-life RAS-based interventions, whether applied individually or in combination, remain incomplete and challenging to predict. Consequently, future efforts should focus on developing optimal methodologies to gain a more holistic understanding of the RAS, ensuring that RAS-based therapy is directed appropriately. Moreover, attention must be given to determining the optimal dosage in a sex-dependent manner to maximize benefits without increasing toxicity before clinical translation.

Even with substantial progress in the accessibility of various RAS-based drugs, there remains a lack of in-depth explorations into their reprogramming effects on each component of CKM syndrome. Another challenge lies in identifying specific developmental windows for different RAS-based therapies to reprogram the processes driving distinct CKM phenotypes, which still await further clarification. Nonetheless, this review marks progress by establishing a connection between the RAS and the developmental origins of CKM syndrome. It provides valuable insights that could pave the way for potential RAS-based interventions aimed at mitigating the global burden of CKM syndrome in the future.

## Figures and Tables

**Figure 1 ijms-25-03298-f001:**
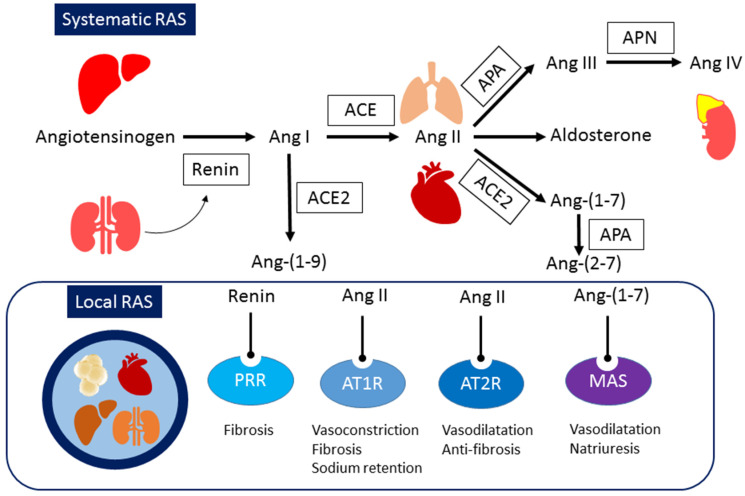
Schema outlining the major organs and components of the renin–angiotensin system.

**Figure 2 ijms-25-03298-f002:**
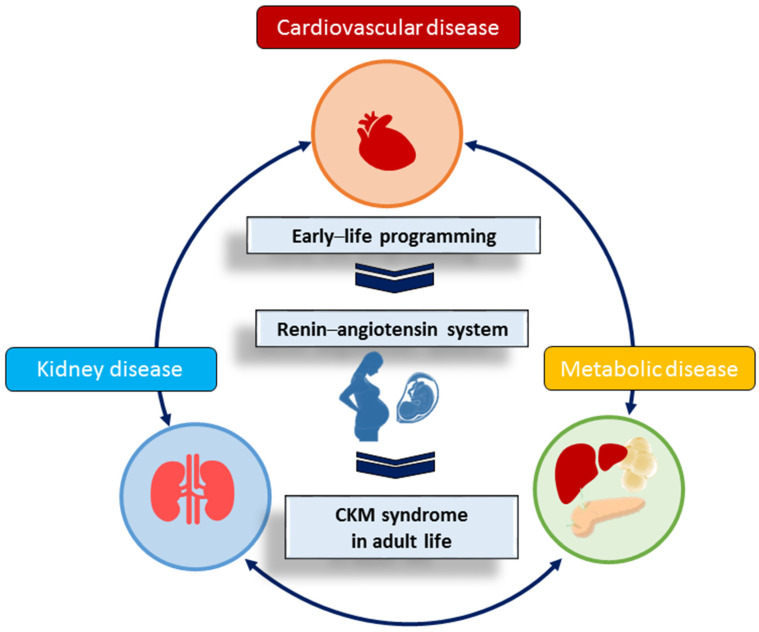
An overview of the role of the renin–angiotensin system in the developmental programming of adult cardiovascular–kidney–metabolic (CKM) syndrome.

**Table 1 ijms-25-03298-t001:** Overview of rat animal models of programmed CKM syndrome related to the aberrant RAS.

Experimental Model	Early-Life Exposure	CKM Phenotype	References
Maternal nutritional imbalance	Caloric restriction	Hypertension, insulin resistance, and kidney disease	[92,93,94]
	Protein restriction	Hypertension, insulin resistance, and kidney disease	[95,96,97,98]
	High-fructose diet	Hypertension, insulin resistance, obesity, and dyslipidemia	[99,100,101,102,103]
	High-fat diet	Hypertension, insulin resistance, obesity, dyslipidemia, and kidney disease	[104,105,106,107]
Maternal illnesses and conditions	Maternal diabetes	Hypertension, insulin resistance, obesity, dyslipidemia, and kidney disease	[108,109,110]
	Maternal chronic kidney disease	Hypertension and kidney disease	[111,112]
	Uteroplacental insufficiency	Hypertension, dyslipidemia, insulin resistance, and kidney disease	[83,113,114,115]
	Maternal hypoxia	Obesity and hypertension	[116,117]
Drug and chemical exposures	Prenatal glucocorticoid exposure	Hypertension, obesity, insulin resistance, and kidney disease	[92,118,119,120,121]
	Prenatal nicotine exposure	Hypertension, hyperlipidemia, steatosis, and kidney disease	[122,123,124,125]
	Prenatal ethanol exposure	Hypertension, insulin resistance, and kidney disease	[126,127]
	Maternal TCDD exposure	Hypertension, cardiac hypertrophy, and kidney disease	[128,129]
	Maternal DEHP exposure	Hypertension, insulin resistance, and kidney disease	[130,131,132,133]

**Table 2 ijms-25-03298-t002:** Interventions targeting the RAS to prevent CKM phenotypes.

Intervention	Experimental Model	Species	Age at Evaluation (Weeks)	Protective Effects	Ref.
Renin inhibitor					
Administration of aliskiren at doses of 10 or 30 mg/kg/day between the ages of 4 and 10 weeks	Genetic hypertension model	SHR/M	10	Hypertension was prevented	[141]
Administration of aliskiren at a dosage of 10 mg/kg/day between the ages of 2 and 4 weeks	Maternal caloric restriction	SD rat/M	12	Hypertension was prevented	[142]
Administration of aliskiren at a dosage of 10 mg/kg/day between the ages of 2 and 4 weeks	Maternal high-fructose diet	SD rat/M & F	12	Hypertension was prevented	[101]
Aliskiren was administered at a dosage of 10 mg/kg/day using a pump from postnatal days 12 to 18	STZ-induced diabetes	TGR (mREN)27 rat/M	16	Diabetic retinopathy was prevented and hypertension was attenuated	[143]
ACEI					
Administration of captopril at a dosage of 100 mg/kg/day between the ages of 2 and 4 weeks	Maternal protein restriction	Wistar rat/M	12	Hypertension was prevented	[144]
Administration of captopril at a dosage of 100 mg/kg/day between the ages of 4 and 10 weeks	Genetic hypertension model	SHR/M	30	Hypertension was attenuated	[145]
Enalapril was administered at a concentration of 100 mg/L in the drinking water between the ages of 3 and 6 weeks	Maternal protein restriction	SD rat/M	16	Hypertension was prevented	[146]
Enalapril was administered at a concentration of 100 mg/L in the drinking water between the ages of 3 and 6 weeks	Maternal protein restriction	SD rat/M	24	Hypertension and albuminuria were prevented	[147]
Lisinopril was administered at a dosage of 10 mg/kg/day through the drinking water from postnatal days 12 to 18	STZ-induced diabetes	TGR (mREN)27 rat/M	16	Hypertension was prevented and diabetic retinopathy was attenuated	[143]
Perindopril was administered at a dosage of 3 mg/kg/day between the ages of 4 and 16 weeks	Genetic hypertension model	SHR/M	28	Hypertension and renal dysfunction were attenuated	[148]
ARB					
Losartan was administered at a concentration of 100 mg/L in the drinking water between the ages of 2 and 4 weeks	Maternal protein restriction	Wistar rat/M	12	Hypertension was prevented	[149]
Losartan was administered at a dosage of 20 mg/kg/day between the ages of 2 and 4 weeks	Maternal caloric restriction	SD rat/M	12	Hypertension was prevented	[150]
Losartan was administered at a dosage of 20 mg/kg/day between the ages of 4 and 9 weeks	Genetic hypertension model	SHR/M	10	Hypertension was prevented	[141]
Losartan was administered at a concentration of 30 mg/L in the drinking water between the ages of 5 and 8 weeks	Uteroplacental insufficiency	WKY rat/M	26	Hypertension, vascular dysfunction, and kidney disease were prevented	[151]
AT1R antisense					
AT1R antisense was delivered at 5 days of age	Genetic hypertension model	SHR/M	12	Hypertension was prevented	[152]
ACE2 activator					
Diminazene aceturate was administered in pregnancy	Maternal hypertension	SHR/M	16	Hypertension and kidney fibrosis were attenuated	[153]
ANG-(1-7) was administered in pregnancy	Maternal hypertension	SHR/M	16	Hypertension and kidney fibrosis were attenuated	[153]

## Data Availability

Data are contained within the article.

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
