# Peer review of "The Renin–Angiotensin System and Cardiovascular–Kidney–Metabolic Syndrome: Focus on Early-Life Programming"

_ijms, 2024, doi:10.3390/ijms25063298_

Round 1

Reviewer 1 Report

Comments and Suggestions for Authors

In this Review, the Authors underscore the importance of understanding prenatal risk factors and their impact on Cardiovascular-Kidney-Metabolic (SKM) syndrome across an individual's lifespan, which is a significant area of research with potential implications for public health. Specifically, it emphasizes the role of the renin-angiotensin system (RAS) in maternal-fetal health and in the development of CKM syndrome, providing a mechanistic framework for understanding this condition. The Review provides an overview of interventions targeting RAS to prevent CKM traits, which could be valuable for clinicians and researchers looking for actionable strategies.

This is a strong Review focused on the important problem of the prevention of cardiovascular diseases. This Review presents a lot of analytical findings. It can be published. There are only several small recommendations that the Authors can easily address.

1. 2. Systemic and Local RAS

Please, simplify the description of RAS. The current version is quite confusing even despite the Figure 1. Perhaps, it would be good to prepare one more scheme.

2. At page 11 you wrote:

Investigations into the reprogramming effects of RAS-based treatments have been conducted in rats aged between 10 and 30 weeks, approximately aligning with human ages from childhood to young adulthood.

Please, provide references confirming this notion.

3. It would be good to prepare a small chapter focused on the importance of the early postnatal period. This period covers not only the window of therapeutic opportunities, but also it coincides with one more critical window of heart terminal differentiation. For example, it was recently shown that neonatal gastroenteritis triggered by lactose intolerance causes long-term transcriptome rearrangements and severe cardoimyocyte remodeling in the rat https://doi.org/10.3390/ijms24087063.

Author Response

RESPONSES TO REVIEWER’S COMMENTS

Reviewer #1

In this Review, the Authors underscore the importance of understanding prenatal risk factors and their impact on Cardiovascular-Kidney-Metabolic (SKM) syndrome across an individual's lifespan, which is a significant area of research with potential implications for public health. Specifically, it emphasizes the role of the renin-angiotensin system (RAS) in maternal-fetal health and in the development of CKM syndrome, providing a mechanistic framework for understanding this condition. The Review provides an overview of interventions targeting RAS to prevent CKM traits, which could be valuable for clinicians and researchers looking for actionable strategies.

This is a strong Review focused on the important problem of the prevention of cardiovascular diseases. This Review presents a lot of analytical findings. It can be published. There are only several small recommendations that the Authors can easily address.

RESPONSE: We thank the reviewer #1 for the efforts and the constructive comments on the work. We have incorporated your comments throughout the manuscript accordingly. Please find below a point-by-point response.

  1. 2. Systemic and Local RAS

Please, simplify the description of RAS. The current version is quite confusing even despite the Figure 1. Perhaps, it would be good to prepare one more scheme.

RESPONSE: We have revised this section for clarity as follows. Figure 1 serves as a comprehensive overview of the entire concept presented in this review, tailored for a general readership. Our intention is to retain its current form to offer readers a holistic understanding of the concept.

P3: “Initiating the RAS cascade is renin, with its precursor, prorenin (406 amino acids) [29], undergoing proteolytic cleavage exclusively in the kidney, where both renin and pro-renin are secreted into circulation [30]. Renin (340 amino acids), acting as a hormone [31], binds to (pro)renin receptor (PRR), encoded by Atp6ap2 in three forms [32]. The interaction of circulating renin and prorenin with PRR triggers Ang II-independent signaling cascades, initiating local Ang II generation.

The RAS substrate, AGT, released from the liver, is cleaved by renin to produce Ang I. ACE further cleaves Ang I, leading to Ang II formation in various tissues [33]. While AT1R stimulation by Ang II increases sodium reabsorption and raises BP, AT2R mediates vasodilation and lowers BP [34]. Ang II also promotes lipogenesis [35], increases adipose tissue mass, and stimulates the adrenal gland cortex to secrete aldosterone, maintaining sodium-potassium homeostasis. The renal RAS, with the highest tissue concentrations of ANG II, involves the metabolism of Ang II to Ang III and Ang IV [36].

ACE2 converts Ang II to Ang-(1–7) or Ang I to Ang-(1–9). Ang-(1–7), mediated by the MAS receptor, induces natriuretic and diuretic effects, promoting vasodilation [37]. Nephrilysin (NEP) facilitates the conversion of Ang I to Ang-(1–7), with subsequent metabolic processing generating Ang-(2–7) and Ang-(3–7).

Distinguishing between local and systemic RAS poses challenges due to extensive overlap [38]. The local adipose RAS, expressed in adipose tissues, modulates processes such as adipogenesis, lipogenesis, lipolysis, and inflammation [39]. The kidney houses a potent local vascular RAS for independent renal vascularization. A distinct urinary RAS in the kidney coordinates sodium reabsorption [40].

A comprehensive understanding of the RAS peptide network's influence on fetal programming requires recognizing the collaborative or opposing nature of different pep-tides. Pharmacological modifications induce compensatory adjustments in RAS enzymes, necessitating further research to unravel the complexities of this network and its impact on fetal programming, as illustrated in Figure 1.”

  1. At page 11 you wrote:

Investigations into the reprogramming effects of RAS-based treatments have been conducted in rats aged between 10 and 30 weeks, approximately aligning with human ages from childhood to young adulthood.

Please, provide references confirming this notion.

RESPONSE: We appreciate your valuable comment. To clarify, one rat month is equivalent to three human years, and we have included a reference to support this information.

  1. It would be good to prepare a small chapter focused on the importance of the early postnatal period. This period covers not only the window of therapeutic opportunities, but also it coincides with one more critical window of heart terminal differentiation. For example, it was recently shown that neonatal gastroenteritis triggered by lactose intolerance causes long-term transcriptome rearrangements and severe cardoimyocyte remodeling in the rat https://doi.org/10.3390/ijms24087063.

RESPONSE: We concur with the reviewer's observation regarding the significance of the early postnatal period in developmental programming. In rodents, kidney development is completed by postnatal weeks 1-2, and cardiomyocytes exhibit limited reentry or progression through the cell cycle beyond postnatal day 9. Our review also surveys RAS-targeted reprogramming strategies during this critical phase. Nevertheless, therapeutic RAS-based interventions typically initiate as early as postnatal 2 weeks, as listed in Table 2. We appreciate the reviewer for providing an intriguing paper, and in response, we have incorporated the following statement to underscore this aspect:

P11: "In rodents, kidney development is entirely completed by the postnatal weeks 1-2 period, and cardiomyocytes seldom reenter or advance through the cell cycle after postnatal day 9. Consequently, suitable therapeutic windows entail the initiation of treatments in juvenile offspring, commencing as early as postnatal 2 weeks in most rodent models."

Reviewer 2 Report

Comments and Suggestions for Authors

 First of all, I doubt the creation of the new term - cardiovascular-kidney-metabolic. There are few papers (9) in PubMed using it. It seems inappropriate, especially since the metabolic disturbances related to obesity seem to be the trigger mechanism.  Perhaps we should not create a new entity. There is no need to create new terms. The proposed syndrome is already within the metabolic syndrome term.

The arguments supporting programming related to the intrauterine environment, especially during adulthood it quite low. They are strongly modified by lifestyle. 

The RAS is far from intrauterine programming, especially during adult life. Its activity is hardly related to sodium consumption. Indeed, various molecular mechanisms associated with the developmental programming of CKM syndrome have been explored, but the evidence is weak.

I do not see any argument confirming the causal function of RAS for the cardio-renal consequences of the metabolic syndrome. It is the opposite. RAS activity is related to the factors directly and indirectly related to obesity

In my opinion, the concept of the paper is misleading.

It is not true that endothelial dysfunction is generated by high renin activity. It is just a coincidence.

The knock animal studies cannot be translated to programming changes.

RAS activation on DM2 is mostly secondary to obesity.

Author Response

Reviewer #2

First of all, I doubt the creation of the new term - cardiovascular-kidney-metabolic. There are few papers (9) in PubMed using it. It seems inappropriate, especially since the metabolic disturbances related to obesity seem to be the trigger mechanism.  Perhaps we should not create a new entity. There is no need to create new terms. The proposed syndrome is already within the metabolic syndrome term.

RESPONSE: We express our appreciation to Reviewer #2 for his/her meticulous evaluation and valuable insights into our manuscript.

It is crucial to recognize that this syndrome is not an invention of ours, and its significance has been acknowledged by the scientific community. The recent official recognition of CKM syndrome by the American Heart Association (AHA) through its Scientific Statement since 2023 underscores the timeliness and urgency of our review, adding significant scientific interest. Notably, the AHA's definition emphasizes that CKM syndrome extends beyond metabolic syndrome or obesity alone; either can be indicative of CKM syndrome.

Despite the well-established global disease burden associated with diabetes, obesity, chronic kidney disease (CKD), and CVD, the intricate and mutually reinforcing detrimental relationships among them remain largely unclear. These components share risk factors and can lead to one another. Therefore, the AHA's 2023 Scientific Statement defines CKM syndrome as a systemic disorder characterized by pathophysiological interactions among metabolic risk factors, CKD, and the cardiovascular system. The classification of CKM syndrome into four stages, ranging from stage 0 to stage 4, aims to capture the diverse degrees of progression and severity within this complex spectrum. Different key components manifest at various stages, contributing to the nuanced progression and severity of CKM syndrome.

We sincerely hope that the Reviewer grasps the importance of CKM syndrome and understands that its recognition is a result of broader scientific consensus.

The arguments supporting programming related to the intrauterine environment, especially during adulthood it quite low. They are strongly modified by lifestyle.

RESPONSE: Despite numerous preclinical studies suggesting that an adverse intrauterine environment can elevate long-term cardiovascular, kidney, and metabolic risks in adult offspring, conventional approaches to screening pregnant women and their children are not fully utilized in clinical practice. The impact of lifestyle on the programming process may be acknowledged but requires additional investigation for confirmation. Despite the growing field of research in DOHaD, it is important for the reviewer to recognize that there are already over 2000 papers addressing this issue.

The RAS is far from intrauterine programming, especially during adult life. Its activity is hardly related to sodium consumption. Indeed, various molecular mechanisms associated with the developmental programming of CKM syndrome have been explored, but the evidence is weak.

RESPONSE: The significance of RAS in developmental programming is underscored not only by our perspective but also by the consensus among other researchers. Various reviews have highlighted the role of RAS in specific aspects of CKM syndrome, including hypertension (refer to Hypertension 2023, 80, e75-e89), obesity (see J Nutr Biochem 2023, 113, 109252), and kidney disease (refer to J Pediatr Endocrinol Metab 2023, 36, 615-627). Nonetheless, our review stands out as the inaugural effort to present a comprehensive overview of the influence of RAS on offspring CKM syndrome.

We are concerned about a potential misunderstanding by the reviewer regarding our assertion. It is crucial to clarify that aberrant RAS is linked to the dysregulation of sodium transporters, not sodium consumption. Based on the evidence we have presented; we find it challenging to concur with the suggestion that our evidence is weak. Nevertheless, we welcome additional insights from the knowledgeable reviewer to ensure we have not overlooked any pertinent information.

I do not see any argument confirming the causal function of RAS for the cardio-renal consequences of the metabolic syndrome. It is the opposite. RAS activity is related to the factors directly and indirectly related to obesity

RESPONSE: It is imperative to clarify that CKM syndrome extends beyond metabolic syndrome, and attributing the cardio-renal consequences solely to the metabolic syndrome is an inaccurate interpretation of CKM syndrome. Our review underscores that each component of CKM shares common risk factors, potentially leading to one another. Consequently, our analysis proposes that the RAS serves as a central nexus connecting all facets of CKM syndrome. We remain uncertain whether RAS is exclusively linked to factors directly and indirectly associated with obesity. Notably, in the context of chronic kidney disease, a significant number of patients exhibit a connection to aberrant RAS rather than obesity. We encourage the inclusion of further perspectives from the experienced reviewer to guarantee that we have thoroughly considered all relevant information and have not missed any critical details.

In my opinion, the concept of the paper is misleading.

RESPONSE: We genuinely trust that the Reviewer recognizes the significance of CKM syndrome and comprehends that its acknowledgment is a reflection of a broader scientific consensus. Given that other reviewers also perceive the importance of this concept, we will defer the decision to the editors and readers.  

It is not true that endothelial dysfunction is generated by high renin activity. It is just a coincidence.

RESPONSE: In accordance with the suggested revision, we have rephrased our statement as follows: "Endothelial dysfunction may arise through the activation of PRR, leading to elevated Ang II activity [44]."

The knock animal studies cannot be translated to programming changes.

RESPONSE: We are uncertain about the knockout animal study to which you referred, as none of the animal models listed in Tables 1 and 2 correspond to knockout models.

RAS activation on DM2 is mostly secondary to obesity.

RESPONSE: Our review emphasizes the equal importance of each component within CKM. Each of these components shares common risk factors, which can potentially lead to one another. The RAS acts as a central hub, interconnecting all aspects of CKM syndrome.

Round 2

Reviewer 1 Report

Comments and Suggestions for Authors

The Article can be published. The Authors addressed all my comments.

Author Response

RESPONSES TO REVIEWER’S COMMENTS

Reviewer #1

The Article can be published. The Authors addressed all my comments.

RESPONSE: We would like to extend our gratitude to the Reviewer for his/her thorough assessment and generous assistance.

Reviewer 2 Report

Comments and Suggestions for Authors

I have come true the the paper 'Cardiovascular-Kidney-Metabolic Health: A Presidential Advisory From the American Heart Association'. There is a definition of  CKM syndrome - a health disorder attributable to connections among obesity, diabetes, chronic kidney disease (CKD), and cardiovascular disease (CVD), including heart failure, atrial fibrillation, coronary heart disease, stroke, and peripheral artery disease. CKM syndrome includes those at risk for CVD and those with existing CVD.' 

I will not fight with this term, which is far from the usual meaning of syndromes in medicine that have common, sometimes not fully understood pathogenesis. In this case, it is a set of risk factors strongly related to an unhealthy lifestyle. It is rather a cluster of diseases that develop in middle-aged and older adults increasing the cardiovascular risk. Some roles of RAAS system can be considered.

The systematic review is fair, but the conclusions section is not acceptable. Please tone down a little beat.

'While the dysregulation of the RAS is recognized as a key factor influencing the developmental programming of CKM syndrome, significant gaps persist in the field, primarily due to methodological constraints and a lack of consensus that has impeded translation into clinical practice.'

I cannot accept that is 'a key factor' rather 'one of factors' not 'programming CKM syndrome' but rather 'programming components of CKM' 

Please add that modulation of RAAS by using the drugs is more that fetal reprogramming.

Please change in abstract: potential role of RAS in early-life programming of CKM syndrome

Author Response

Reviewer #2

I have come true the the paper 'Cardiovascular-Kidney-Metabolic Health: A Presidential Advisory From the American Heart Association'. There is a definition of CKM syndrome - a health disorder attributable to connections among obesity, diabetes, chronic kidney disease (CKD), and cardiovascular disease (CVD), including heart failure, atrial fibrillation, coronary heart disease, stroke, and peripheral artery disease. CKM syndrome includes those at risk for CVD and those with existing CVD.'

I will not fight with this term, which is far from the usual meaning of syndromes in medicine that have common, sometimes not fully understood pathogenesis. In this case, it is a set of risk factors strongly related to an unhealthy lifestyle. It is rather a cluster of diseases that develop in middle-aged and older adults increasing the cardiovascular risk. Some roles of RAAS system can be considered.

The systematic review is fair, but the conclusions section is not acceptable. Please tone down a little beat.

RESPONSE: Again, we want to convey our gratitude to Reviewer #2 for careful evaluation and valuable insights into our manuscript. In particular, the reviewer concurs with us regarding the significance of CKM syndrome.

'While the dysregulation of the RAS is recognized as a key factor influencing the developmental programming of CKM syndrome, significant gaps persist in the field, primarily due to methodological constraints and a lack of consensus that has impeded translation into clinical practice.'

I cannot accept that is 'a key factor' rather 'one of factors' not 'programming CKM syndrome' but rather 'programming components of CKM'

RESPONSE: We value your perspective regarding the characterization of the RAS as 'a key factor' in the development of CKM syndrome. We acknowledge your suggestion that it might be more accurate to describe it as 'one of the factors' contributing to the programming of components within CKM syndrome. Your feedback has been duly noted, and we have taken it into consideration, resulting in a revised conclusion as follows. Thank you for sharing your insights.

“While the dysregulation of the RAS is recognized as one of the factors contributing to the programming of components within CKM syndrome, significant gaps persist in the field, primarily due to methodological constraints and a lack of consensus that has impeded translation into clinical practice.”

Please add that modulation of RAAS by using the drugs is more that fetal reprogramming.

RESPONSE: In response to the suggestion, we have incorporated the following statement into the Conclusion: "The utilization of drugs to modulate the RAS is well-established in clinical practice, although it is still emerging in the field of fetal programming."

Please change in abstract: potential role of RAS in early-life programming of CKM syndrome

RESPONSE: We have made the necessary adjustments to our abstract in line with your feedback.
